# A Comparative Longitudinal Study Analyzing Vaginal Microbiota Differences Between Term and Preterm Pregnancies in Korean Women

**DOI:** 10.3390/medicina61040752

**Published:** 2025-04-18

**Authors:** Gina Nam, Kyung A. Lee, Soo Jung Kim, Kwan Young Oh, Sunghee Lee, Hyun Chul Lee, So Yoon Kim, Mi Hye Park

**Affiliations:** 1Department of Obstetrics and Gynecology, Chung-Ang University Hospital, Chung-Ang University College of Medicine, Seoul 06973, Republic of Korea; ginanam@cau.ac.kr; 2Department of Obstetrics and Gynecology, Ewha Medical Center, Ewha Womans University College of Medicine, Seoul 07804, Republic of Korea; leekyunga@gmail.com (K.A.L.); bossksj25@gmail.com (S.J.K.); 3Department of Obstetrics and Gynecology, Eulji University Hospital, Daejeon 35233, Republic of Korea; ohky5@eulji.ac.kr1; 4Daygen Inc., Seoul 06739, Republic of Korea; slee@daygen.co.kr; 5Research Center, D&P Biotech Inc., Seoul 05855, Republic of Korea; hclee@world-medi.com (H.C.L.); soyoon@world-medi.com (S.Y.K.)

**Keywords:** vaginal microbiota, preterm birth, cervical length, longitudinal study

## Abstract

*Background and Objectives*: Preterm birth (PTB), defined as delivery before 37 weeks of gestation, remains a significant public health concern due to its association with neonatal morbidity and mortality. Although studies have suggested that microbial factors in vaginal microbiota (VMB) influence PTB, longitudinal research on Korean women is limited. This study aimed to analyze VMB differences between term and preterm pregnancies in Korean women and their correlation with the cervical length (CL). *Materials and Methods*: A cohort of 60 pregnant Korean women (40 who had a term birth (TB) and 20 who had a PTB) was recruited. Vaginal samples were collected at five time points (first, second, and third trimester; 1–2 weeks postpartum; 1–2 months postpartum). Microbial DNA was extracted and analyzed using quantitative PCR targeting 12 bacterial species. The CL was measured in the second and third trimesters. *Results*: *Lactobacillus crispatus* was consistently dominant in the TB group, whereas PTB cases exhibited greater microbial diversity with elevated levels of *Prevotella salivae* and *Ureaplasma* species. The CL was significantly shorter in PTB cases, correlating with shifts in the VMB composition. *Conclusions*: A stable, *Lactobacillus*-dominant microbiome is protective in pregnancy, while increased diversity in PTB cases suggests microbial biomarkers for early risk prediction. Combining VMB profiling with CL measurement may enhance early, non-invasive PTB risk assessments.

## 1. Introduction

Preterm birth (PTB) occurs before 37 weeks of gestation and poses significant health risks, including neonatal morbidity and mortality [1,2,3]. Global PTB rates are rising, with substantial healthcare and socioeconomic costs [4,5,6]. PTB is associated with various risk factors, including maternal age, ethnicity, smoking, and infections such as bacterial vaginosis and chorioamnionitis [7,8]. Infection-related PTB is linked to the microbial invasion of the amniotic cavity (MIAC), often involving *Ureaplasma*, *Gardnerella*, and *Fusobacterium* species [9]. The vaginal microbiota (VMB) undergoes changes during pregnancy, with *Lactobacillus* dominance associated with term birth, while increased microbial diversity and dysbiosis elevate the PTB risk [10,11]. The cervical length (CL) is a key clinical marker, with a shorter CL correlating with a higher PTB risk [12,13]. Traditional PTB diagnostics rely on clinical symptoms and ultrasound, but emerging molecular techniques now allow for comprehensive microbial analysis. This study aimed to analyze VMB differences between term and preterm pregnancies in Korean women and integrate microbial and clinical data for enhanced PTB prediction.

## 2. Materials and Methods

### 2.1. Patient Selection and Vaginal Sample Collection

A cohort of healthy singleton pregnancies was prospectively enrolled at Ewha Womans University Seoul Hospital from March 2022 to January 2024 to investigate the relationship between the VMB, CL, and PTB. Ethical approval was obtained (IRB Nos. 2021-12-037 and 2023-09-029), and all participants provided written consent. Based on previous studies of the VMB in pregnant Korean women, 12 key bacterial species were analyzed across four CSTs, including *Lactobacillus crispatus*, *L. iners*, *L. gasseri*, *L. jensenii*, *Weissella koreensis*, *Ureaplasma urealyticum*, *U. parvum*, *Gardnerella vaginalis*, *Bacteroides fragilis*, *Prevotella bivia*, *P. salivae*, and *P. amnii*. A total of 695 singleton pregnancies were initially recruited, comprising 609 term deliveries and 86 preterm deliveries. In the term birth (TB) group, 337 participants were excluded due to incomplete sample collection, while an additional 169 were excluded based on medical conditions such as fetal abnormalities, gestational diabetes, hypertensive disorder, kidney disease, thyroid disease, other significant medical conditions, or antibiotic, antifungal, and vaginal progesterone use one week before vaginal fluid collection. After the exclusions, 103 participants from the TB group were randomized, resulting in 40 patients with study outcomes. In the PTB group, 39 participants were excluded due to insufficient sampling, and an additional 27 were excluded based on similar medical conditions, resulting in 20 participants being enrolled. Only participants with complete sample collection at five time points in the TB group and three time points in the PTB group were included. Ultimately, 60 participants were included in the study: 40 from the TB group and 20 from the PTB group (Figure 1). Given that the qPCR microbial data did not follow a normal distribution and were log-transformed prior to analysis, we performed a post hoc power analysis using G*Power (v3.1.9.4). To reflect the characteristics of the log-transformed, skewed data, we assumed a Laplace-like parent distribution. Using a two-sided test with α = 0.05 and a moderate-to-large effect size (Cohen’s d = 0.7), the achieved power was calculated to be 0.872. This result suggests that the sample size used (TB group = 40, PTB group = 20) was adequate for detecting meaningful differences between groups. CVF samples were collected at five time points (first, second, and third trimester; 1–2 weeks and 1–2 months postpartum) using sterile cotton swabs (E-swab, Copan, Italy). The swabs were mixed with Liquid Amies Medium, placed on ice, and stored at −80 °C within 5 min. The CL was measured during the second and third trimesters using transvaginal ultrasonography (TVS), with three measurements taken per session and averaged for accuracy. Medical records were reviewed to obtain participants’ demographic data, obstetric history, and pregnancy outcomes, including the gestational age at birth and delivery method. Routine blood tests, including measurements of the white blood cell count (WBC) and C-reactive protein (CRP), were performed during the second trimester. Following delivery, pregnancy outcomes were assessed, including the gestational age (GA) at birth, the delivery method, and the indication for delivery.

### 2.2. DNA Extraction and Probe Design

#### 2.2.1. DNA Extraction

Microbial DNA from CVF samples was extracted using the QIAamp^®^ DNA Mini Kit (Qiagen, Hilden, Germany), following the manufacturer’s instructions, under stringent conditions to minimize potential DNA contamination and degradation. CVF pellets were obtained by centrifugation at 7500 rpm for 10 min, resuspended in 1 mL of phosphate-buffered saline (PBS), and gently mixed for 30 s to ensure sample homogeneity. Prior to DNA extraction, 10 μL of an internal control was added to each sample to monitor the efficiency and consistency of the extraction process. The extraction procedure included the addition of 20 μL of Proteinase K (20 mg/mL) and 400 μL of an AL buffer (Qiagen) to 370 μL of the sample suspension, followed by mixing for 15 s and incubation at 56 °C for 10 min to ensure complete lysis while preserving DNA integrity. Subsequently, 400 μL of ethanol (96–100%) was added, and the mixture was carefully transferred to Qiagen spin columns to allow for high-purity DNA binding and purification. The elution step was carried out using 100 μL of an AE buffer (Qiagen), and the extracted DNA was immediately quantified using a NanoDrop ND-2000 spectrophotometer (Thermo Fisher Scientific, Waltham, MA, USA). To prevent degradation, all DNA samples were handled on ice during processing and promptly stored at −80 °C until further analysis. Throughout the workflow, we employed sterile consumables, carried out procedures in a DNA-free workspace, and followed best practices for nucleic acid handling to ensure sample integrity and prevent cross-contamination.

#### 2.2.2. Primer and Probe Design

For the primer and probe design, the nucleotide sequences of 12 target bacterial strains were obtained from NCBI RefSeq and converted to the FASTA format. Using Gene Runner software (V6.5.2) and BLAST alignment, primers and probes were designed for species-specific genes, with parameters including a 21–27 nucleotide length, 29–55% guanine–cytosine content, and a melting temperature (Tm) of 57–60 °C. The targeted bacteria included *L. crispatus*, *L. iners*, *L. gasseri*, *L. jensenii*, *W. koreensis*, *U. urealyticum*, *U. parvum*, *G. vaginalis*, *B. fragilis*, *P. bivia*, *P. salivae*, and *P. amnii*. The Arabidopsis Ceas3 gene was used as an internal control to enhance the reliability of the qPCR assay. Information on the designed primers and probes is shown in Appendix A. Species-specific primers and probes were labeled with FAM, CY5, JOE, or Texas Red fluorescence and synthesized commercially by Cosmogenetech (Seoul, Republic of Korea) and Bionics (Seoul, Republic of Korea).

### 2.3. Target Plasmid Preparation for Standard Curve Production

A standard curve was produced by performing conventional PCR using DNA extracted from a standard strain. A total volume of 20 µL was used for the amplification, containing 10 µL of HelixAmp Ready-2X-Go (NanoHelix, Daejeon, Republic of Korea), 5 µL of an oligo mix (10 pmole/rxn), 1 µL of DNA (1 ng/rxn), and 4 µL of a TE buffer (Bioneer, Daejeon, Republic of Korea). The PCR reactions began with an initial denaturation step at 95 °C for 15 min, followed by 42 cycles of 95 °C for 10 s and 60 °C for 1 min. The PCR amplification products were purified using a FavorPrep GEL/PCR Purification Mini Kit (Favorgen, Pingtung, Taiwan). Each of the 12 targeted genes was cloned into a PCR^®^ 2.1-TOPO^®^ vector (Invitrogen, Waltham, MA, USA) or pGEM T-Vector (Promega, Madison, WI, USA). The eight specific strains (*L. crispatus*, *W. koreensis*, *P. bivia*, *P. amnii*, *P. salivae, B. fragilis*, *U. urealyticum*, and *U. parvum*) were cloned using the pGEM T-Vector (Promega, Madison, WI, USA), while the PCR 2.1-TOPO vector (Invitrogen, Carlsbad, CA, USA) was used for cloning the remaining four targeted strains (*L. iners*, *L. jensenii*, *L. gasseri*, and *G. vaginalis*). The produced plasmid DNA was purified using a GeneAll^®^ Exprep™ Plasmid SV Mini kit (GeneAll Biotechnology, Seoul, Republic of Korea). The concentration and yield of the plasmid DNA were quantified using a NanoDrop ND-2000 spectrophotometer (Thermo Fisher Scientific, Wilmington, DE, USA) and diluted to a concentration of 1 ng/µL. Standard positive controls were prepared so that each target gene was present at a concentration of 10^7^ copies/5 µL and were diluted 10-fold to 10^3^ copies/5 µL.

### 2.4. Optimization of Simplex and Multiplex qPCR Assays

#### 2.4.1. Specificity and Accuracy of qPCR Assays

The performance of the oligo sets was verified using standard strains purchased from the American Type Culture Collection (ATCC), Japan Collection of Microorganisms (JCM), DSMZ-German Collection of Microorganisms (DSM), and Korean Agricultural Culture Collection (KACC). Forty-one related and non-related strains that could possibly exist in the same sample as the twelve targeted strains were selected (Appendix A). For specificity testing, the DNA concentration of each strain was normalized to 1 ng/rxn, and as a result of the test, it was confirmed that no cross-reaction occurred in other strains except for the 12 targeted strains, and the specificity for the target strains was confirmed. For accuracy testing, a standard curve was established using all standard positive controls over the range of 10^3^–10^7^ copies/5 uL. For primers with high efficiency, a standard curve must exhibit an R^2^ value ≥ 0.98 and a slope ranging from −3.1 to 3.6 [14]. A combination of multiplex assays consisting of 4 oligo sets was used to analyze the 12 targeted strains. The amplification curves obtained from the qPCR reaction for the 12 targeted strains were good. The standard curves of Oligo Set A (*L. crispatus*, *W. koreensis*, and *L. iners*) had R^2^ values of 0.999, 0.999, and 0.998, showed slopes of −3.359, −3.331, and −3.399, and showed amplification efficiencies of 98.484%, 99.627%, and 96.865%, respectively. Oligo Set B (*U. urealyticum*, *U. parvum*, and *G. vaginalis*) had R^2^ values of 0.999, 0.999, and 0.998, showed slopes of −3.355, −3.524, and −3.504, and showed amplification efficiencies of 98.622%, 92.214%, and 92.943%, respectively. Oligo Set C (*L. gasseri*, *L. jensenii*, and *B. fragilis*) had R^2^ values of 0.999, 0.998, and 0.998, showed slopes of −3.342, −3.256, and −3.273, and showed amplification efficiencies of 99.186%, 102.844%, and 102.07%, respectively. Oligo Set D (*P. bivia*, *P. salivae*, and *P. amnii*) had R^2^ values of 0.993, 0.997, and 0.997, showed slopes of −3.319, −3.356, and −3.418, and showed amplification efficiencies of 101.106%, 98.614%, and 96.149%, respectively (Appendix A). This indicates that 4 oligo sets exhibited high efficiency and specificity for distinguishing the target species in a sample.

#### 2.4.2. Sensitivity of qPCR Assays

Detection sensitivity tests of the multiplex assays of 12 targeted strains were performed using 4 concentrations of diluted standard positive controls, such as 10, 50, 100, and 1000 copies/rxn, and a TE buffer was used as a negative control with 0 copies. As a result of performing repeated tests, the concentration where more than 95% of the samples were detected was determined as the limit of detection (LoD), and performance above the detection limit of 100 copies/rxn was confirmed.

### 2.5. Quantification of Vaginal Bacteria by Real-Time PCR (qPCR)

The absolute quantification of 12 targeted strains was performed using an AB7500 instrument and 7500 software, v2.3 (Applied Biosystems, Foster City, CA, USA). A combination of multiplex assays consisting of 4 oligo sets was used to analyze the 12 targeted strains. These were fourplex qPCR assays consisting of three target microorganisms and one internal control. For qPCR, a final volume of 20 μL comprised 10 µL of the RealHelix™ Superplex qPCR Kit (NanoHelix, Daejeon, Republic of Korea), 5 µL of the respective primers and probes, and 5 µL of the template DNA. Initial denaturation at 95 °C for 15 min was followed by 42 cycles of 95 °C for 10 s and 60 °C for 1 min during the amplification process. A positive control using plasmid DNA for each targeted gene and a negative control using a TE buffer were included throughout the procedure. The standard curve was analyzed for three technical replicates. Five different dilutions (10^7^ to 10^3^) of the standard positive controls were used, whose concentrations had already been calculated. The slope, R^2^, and efficiency were calculated by plotting the Ct value against the log starting value of the standard positive controls. After obtaining the copy numbers, the relative quantity of each of the 12 targeted strains was determined for each subject. The quantity of bacteria was calculated based on the copy number per reaction, which was used to determine the percentage of the target bacteria relative to the total number of detected bacteria. The relationship between the Ct value and the logarithm of the initial copy number was confirmed, with a correlation coefficient value of 0.993 or higher for all targets, and all targets also showed a 90–110% PCR efficiency.

### 2.6. Statistical Analyses

Differences in clinical characteristics between the TB and PTB groups were analyzed using Chi-square tests for categorical variables, while continuous variables were reported as means (standard deviation) or the median (range) and compared using the Mann–Whitney test. We utilized Pearson’s correlation test to evaluate the relationships between continuous variables that followed a normal distribution. The correlation coefficients (rho) were categorized as follows: 0 to 0.1 indicated a “very weak” association, 0.1 to 0.3 a “weak” association, 0.3 to 0.7 a “moderate” association, and 0.7 to 1.0 a “strong” association. Statistical significance was defined as a *p*-value less than 0.05. Statistical analyses were conducted using SPSS (version 20.0) (Chicago, IL, USA) and Python (ver. 3.13.0).

## 3. Results

### 3.1. The Characteristics of the Study Participants

Table 1 shows a detailed comparison of the characteristics between the TB (*n* = 40) and PTB (*n* = 20) groups. The average age of participants was slightly higher in the PTB group (34.4 ± 3.8 years) compared to the TB group (33.0 ± 3.8 years), but this difference was not statistically significant (*p* = 0.191). The pre-pregnancy BMI was also higher in the PTB group, with an average of 24.8 ± 5.9 kg/m^2^, compared to 21.9 ± 3.1 kg/m^2^ in the TB group, though this difference was also not significant (*p* = 0.119). In terms of parity, both groups were relatively similar: 50% of TBs and 65% of PTBs occurred in nulli-parity, with no significant differences across parities (*p* = 0.296). Similarly, a history of previous PTBs was rare in both groups, with only 2.5% of TB and 5.0% of PTB participants reporting prior preterm deliveries, a non-significant difference (*p* = 0.611). The method of conception revealed a trend towards higher ART use among the PTB group (35.0%) compared to the TB group (20.0%), though this difference did not reach statistical significance (*p* = 0.206). The CL was notably shorter in the PTB group across both the second and third trimesters. In the second trimester, the average CL was 43.0 ± 6.0 mm for the TB group and 38.3 ± 8.1 mm for the PTB group (*p* = 0.005). By the third trimester, this difference was even more pronounced, with the TB group averaging 30.9 ± 9.1 mm, compared to only 19.7 ± 11.2 mm in the PTB group (*p* = 0.009). The gestational age at delivery was significantly different between the groups, with TBs occurring at a median of 38.5 weeks (range: 38.0–39.4 weeks) and PTBs at a median of 35.8 weeks (range: 34.2–36.4 weeks; *p* = 0.000). This difference was also reflected in the birth weight, where term infants had a higher mean weight of 3232.5 ± 306.8 g, compared to 2421.5 ± 792.2 g for preterm infants (*p* = 0.000). In terms of neonatal Apgar scores, the 1-min scores were similar between the groups, but a slight difference was observed in the 5-min scores, with the PTB group scoring marginally lower (median of 10.0, range of 8.3–10.0) than the term group (median of 10.0, range of 10.0–10.0), reaching statistical significance (*p* = 0.047). Given the non-parametric distribution of the data, we calculated the effect size using the rank-biserial correlation, which was 0.228, indicating a small effect size and suggesting limited clinical relevance.

### 3.2. Longitudinal Analysis of Vaginal Microbiome in Term and Preterm Birth Groups

The TB and PTB groups were evaluated at five time points during pregnancy and postpartum: T1 (first trimester), T2 (second trimester), T3 (third trimester), P1 (1–2 weeks after delivery), and P2 (1–2 months after delivery). The TB group maintained full sample collection at each time point, reflecting comprehensive data coverage for comparison. In the PTB group, samples were gathered across timepoints T1 (with 4 participants), T2 (19 participants), T3 (17 participants), and two postpartum timepoints, P1 and P2 (20 and 18 participants, respectively). The copy numbers of each species were displayed on a logarithmic scale, allowing us to observe the trends and changes in the VMB composition as the pregnancy progressed and following delivery.

Figure 2a shows the abundance of various species within the VMB of term birth participants across five time points. The results indicate a microbiome profile dominated by *Lactobacillus* species. *L. crispatus* was consistently present at high levels, with copy numbers remaining at around 10^8^ across all stages. Other *Lactobacillus* species, such as *L. jensenii* and *L. iners*, remained moderately abundant, with their highest levels seen in the mid-pregnancy stages (T2 and T3). *W. koreensis* also showed a stable presence during pregnancy and the postpartum period, suggesting that it may contribute to maintaining a balanced microbiome during pregnancy and after delivery. Following delivery, the VMB composition showed some shifts. *Lactobacillus* species exhibited a marked reduction in abundance during the postpartum period, while *P. bivia* and *G. vaginalis* showed an increase in abundance in the postpartum period.

Figure 2b illustrates the abundance of various species comprising the VMB of PTB participants across five time points. In the PTB group, *L. crispatus* was also present at relatively high levels during pregnancy, with copy numbers of around 10^7^ to 10^8^. Other *Lactobacillus* species, such as *L. jensenii*, *L. gasseri*, and *L. iners*, were present at moderate levels during pregnancy, a similar trend to that observed in the TB group. In the PTB group, *Prevotella* species showed an increase in abundance during the postpartum period (P1 and P2). Specifically, *P. bivia* demonstrated notable increases in copy numbers after delivery.

### 3.3. Difference in Vaginal Microbiome Between Term and Preterm Birth Groups

Figure 3 shows the differences in the VMB composition between the term and preterm birth groups across all stages of pregnancy and postpartum. In the TB group, three species—*L. crispatus*, *W. koreensis*, and *P. bivia*—were present at consistently higher levels than in the PTB group, with statistically significant differences at various time points. *L. crispatus* was significantly more abundant in the term group, particularly in T2. *W. koreensis* also showed a higher abundance in the TB group across multiple periods, with significant differences at T2, T3, P1, and P2. *P. bivia* was present at higher levels in the TB group during the postpartum period, particularly at P1, where it was significantly more abundant than in the PTB group. In contrast, in the PTB group, *P. salivae* was present at markedly higher levels throughout all periods, with statistically significant differences at each time point (T1, T2, T3, P1, and P2). *U. urealyticum* and *U. parvum* were also more abundant in the PTB group than in the TB group, though their increased levels were primarily observed in the postpartum period (P1 and P2).

### 3.4. Correlation Analysis of Vaginal Microbiome and Cervical Length During Pregnancy

Table 2 shows the correlations between prevalent bacterial species and the CL in the T2 and T3 periods of the PTB group. In the second trimester, *B. fragilis* showed a significant negative correlation with the CL (ρ = −0.542, *p* < 0.01), suggesting a moderate correlation with a shortened CL in the second trimester. *B. fragilis* had a non-significant moderate negative correlation with the CL (ρ = −0.305, *p* = 0.32) in the third trimester. *P. salivae* demonstrated significant correlations with the CL (ρ = −0.693, *p* = 0.03), indicating a moderate negative relationship in the third trimester.

## 4. Discussion

The vaginal microenvironment is influenced by various factors and undergoes significant changes during pregnancy. In the past, studies of vaginal microflora primarily relied on bacterial culture methods. However, these techniques were limited because many microorganisms could not be detected due to the specific conditions required for growth [15]. Many bacteria remain uncultured and unidentified because only a small fraction can grow and form colonies on agar plates under standard laboratory conditions. To address these limitations, culture-independent techniques have been developed and provide a more comprehensive view of microbial diversity and have been successfully applied in numerous studies exploring the vaginal microbiome [16].

The majority of previous studies have concentrated on comparing the vaginal microflora in healthy women with the microflora of those affected by many complications, including infections. The vaginal microbiome is primarily composed of lactic acid-producing *Lactobacillus* species (*L. crispatus*, *L. iners*, *L. gasseri*, and *L. jensenii*), which maintain a low vaginal pH (around 4.5), essential for protecting against infections. BV is characterized by a shift from a *Lactobacillus*-dominated microbiome to a diverse polymicrobial environment, including *G. vaginalis*, *Atopobium vaginae*, *Prevotella*, and other anaerobes. BV can be associated with increased risks of gynecological and obstetric complications, such as PTB, infertility, and pelvic inflammatory disease [17]. Lactic acid plays a vital role in preserving vaginal health, with *Lactobacillus* species suppressing the production of bacteriocins and toxins [18].

Their abundance is enhanced during pregnancy as estrogen levels rise, leading to notable differences in the vaginal microbial composition between pregnant and non-pregnant women within the same age group [19]. Hormones like estrogen and progesterone play a role by increasing the glycogen availability, promoting *Lactobacillus* colonization and leading to a healthier and more stable vaginal environment [20]. Serrano et al. [21] reported that *L. crispatus* remains the predominant species during pregnancy. The VMB also shows an increase in *Lactobacillus* species, particularly *L. iners*, leading to enhanced stability and reduced levels of bacteria associated with BV [19,21]. There have been longitudinal studies on VMB changes during pregnancy across different trimesters [22]. Early stages of pregnancy tend to be associated with a less stable microbial community than later stages. *Lactobacillus* is present as well as some anaerobic bacteria, such as *Gardnerella* and *Prevotella*. In the second trimester, there is an increase in the dominance of *Lactobacillus* and particularly species like *L. crispatus* and *L. iners*, while the presence of BV-associated bacteria like *Atopobium* and *Sneathia* decreases. By the final trimester, the VMB is predominantly stable and largely dominated by *Lactobacillus* species, with a marked reduction in diversity and potentially harmful bacteria. This period represents the most stable microbial state, which is essential for protecting against infections and supporting pregnancy health. After delivery, during the puerperium, the microbiome often shifts back to a more diverse state with decreased *Lactobacillus* dominance and the increased presence of other bacteria like *Gardnerella* and *Prevotella* [22,23]. Kim et al. [24] utilized sequencing techniques to longitudinally examine the composition of the vaginal microbiota across different trimesters of pregnancy in Republic of Korea. In a study examining the VMB of pregnant Korean women who delivered at term, *Lactobacillus* species were found to be predominant across all trimesters. Specifically, *L. crispatus* was the most prevalent, followed by *L. iners*, *L. gasseri*, and *L. jensenii*, with the dominance of *Lactobacillus* species supporting a stable and protective environment throughout pregnancy. Notably, after term delivery, the levels of *Lactobacillus* species, except *L. iners*, dropped significantly, allowing other anaerobes, such as *G. vaginalis* and *B. fragilis*, to become more prevalent postpartum. These shifts suggest that *Lactobacillus*-dominated microbiomes are beneficial during pregnancy, while the postpartum period shows a natural increase in diversity, including anaerobes.

This study showed similar results to those of previous studies. The VMB of the TB group was characterized by the dominance of *Lactobacillus* species, particularly *L. crispatus*. Moderate levels of *L. iners*, *L. jensenii*, and *W. koreensis* further contributed to the composition of the VMB throughout pregnancy. Following delivery, the VMB composition showed shifts, with pathogenic bacteria like *G. vaginalis* and *P. bivia* increasing in abundance postpartum. In the PTB group, *Lactobacillus* species maintained dominance within the microbial community during pregnancy but exhibited a marked reduction in abundance during the postpartum period, similarly to in the TB group. Other *Lactobacillus* species, such as *L. jensenii* and *L. iners*, were observed at moderate levels, following a similar trend to that in the TB group. In contrast, *Prevotella* species demonstrated an increase in abundance during the postpartum period, suggesting a shift in the microbial composition that could reflect postpartum physiological changes within the VMB. This report emphasizes the importance of longitudinal assessments of the vaginal microbiota during pregnancy and the postpartum period, recognizing how microbial dynamics might change across trimesters. It also highlights that further research is essential to understand the mechanisms linking the VMB to pregnancy outcomes like PTB, which could aid in developing targeted interventions for maintaining vaginal health during pregnancy and puerperium.

Additionally, many studies have explored distinctions in the microbial composition between TB and PTB groups, aiming to identify microbial patterns that could predict adverse pregnancy outcomes. During pregnancy, the microbiota plays a crucial role in infection prevention, with its composition adjusting to hormonal changes that promote the increased dominance of *Lactobacilli* [21]. However, the disruption of this balance may impact pregnancy outcomes. Dysbiosis, marked by decreased levels of *L. crispatus* and higher levels of pathogens, is linked to negative outcomes such as PTB [25]. Specific bacterial communities, such as those dominated by *G. vaginalis*, were more associated with PTB than those dominated by *L. crispatus* [26]. Individuals who had a PTB also often had higher microbial diversity in their vaginal microbiota compared to those who had a TB [27]. Additionally, Kumar et al. [28] identified a predictive microbiota for PTB in Asian women, present as early as the first trimester. This microbiota featured elevated levels of *Prevotella buccalis* alongside reduced levels of *L. crispatus* and *L. iners*. These patterns suggest that shifts away from *Lactobacillus* dominance toward pathogenic communities could provide early indications of a PTB risk.

*Weissella* species are a type of lactic acid bacteria recently classified as a distinct genus [29]. They naturally occur in various fermented foods, including traditional Korean fermented vegetables like Kimchi, showing greater resilience in acidic and anaerobic environments. In studies with *Weissella* species, the *Weissella* abundance appeared to be linked to delayed delivery, which has been shown to modify ROS levels and reduce oxidative stress [30]. The stable presence of *Weissella* in individuals who had a TB suggests that it may contribute to a balanced microbiome that protects against preterm labor. In this study, *W. koreensis* was more abundant in individuals who had a TB across the second and third trimesters and postpartum. However, this observation remains speculative, as the current evidence is insufficient to confirm a protective role for *W. koreensis*. Therefore, these findings should be interpreted with caution, and further mechanistic studies are warranted to validate this species’ functional significance.

*Prevotella* species, anaerobic Gram-negative rods, are implicated in the pathogenesis of multiple BV [31]. *Prevotella* genus are also linked to higher risks of PTB, particularly due to their role in inflammatory pathways [32]. *P. bivia*, *P. amnii*, and *P. timonensis* are implicated in female genital tract infections, contributing to biofilm formation, mucosal inflammation, and antibiotic resistance [31]. Studies have indicated that an imbalance in *Prevotella* species can increase the levels of inflammatory mediators that may trigger early labor. However, in this study, it was found that *P. salivae* was consistently more prevalent in the PTB group throughout all stages, whereas *P. bivia* was significantly more abundant in the TB group during the P1 period. The observation that *L. gasseri* showed a higher abundance in the PTB group during the T2 period is also inconsistent with the findings of previous studies. These findings differ from the results of previous studies; however, this may be attributed to the limited sample size.

The observed differences between the term and preterm groups beginning from the second trimester suggest that VMB changes are present from early pregnancy. This finding implies that strategies for predicting and potentially preventing PTB could be developed based on these early microbial shifts. This study emphasizes the importance of monitoring VMB changes throughout pregnancy as potential indicators for interventions to prevent PTB.

Recent studies have explored predictive models for PTB that integrate clinical indicators (such as the CL, CRP, and WBC count) with VMB profiling to improve the prediction accuracy. As demonstrated in numerous studies showing a significant shortening of the CL in the PTB group, our study also observed a marked difference in the CL between term and PTB groups in the second trimester and an even greater disparity in the third trimester. The significant shortening of the CL observed in the PTB group during both the second and third trimesters suggests that the CL may serve as a valuable biomarker for PTB risk assessment. The early detection of a shortened cervix, particularly during routine second-trimester screening, could be critical for timely interventions, such as progesterone supplementation or cerclage placement, which have been shown to reduce the risk of preterm delivery in high-risk women [33,34,35]. Additionally, the progressive shortening of the CL in PTB cases underscores the importance of the continuous monitoring of the cervical status, especially in individuals with risk factors for PTB.

There has been a study that explored the combined role of the CL and immune markers, particularly CRP, in predicting PTB [36]. It reaffirmed that elevated CRP levels, indicative of systemic inflammation, were associated with a higher PTB risk when coupled with a short CL, particularly one below 25 mm. Kindinger et al. [12] reported that women with a short CL and *L. iners* dominance were more likely to be at risk for PTB, while *L. crispatus* dominance was more stable across gestation, especially in individuals who had a TB [12]. Another study employed multiple machine learning algorithms and used the combined VMB, CL, and WBC count data of 150 Korean women to predict PTBs [13]. This model highlighted the CL as a primary factor, followed by an increased prevalence of *U. parvum* and *Peptoniphilus grossensis* in cases with a shorter CL. This dysbiotic state, marked by a decrease in *L. crispatus* and an increase in potential pathogens, correlated with increased inflammation, which can impact cervical function and potentially trigger early labor [37]. Our research suggests that an abundance of *B. fragilis* and *P. salivae* combined with a short CL could serve as a biomarker for predicting PTL. Our study suggests that *P. salivae* levels were significantly higher in the preterm group across all trimesters and even postpartum compared to the term group. Adding the CL to this analysis as a clinical factor could enhance the predictive accuracy for PTB, potentially making the abundance of *P. salivae* combined with the CL a stronger biomarker for identifying PTB risks. This combined approach suggests that the VMB composition in conjunction with CL measurements may serve as a useful predictive marker for PTB. This study presented a promising method for early PTB prediction by integrating microbial and clinical markers, offering potential for non-invasive clinical application to better manage PTB risks.

A limitation of this study is the small sample size in the preterm group during the first trimester, with only four participants. This limited sample size may have reduced the statistical power for detecting early microbiome changes in preterm cases and highlights the need for further studies with larger first-trimester samples to validate these findings. However, this study has several strengths. It offers comprehensive, longitudinal data on VMB changes across all trimesters and postpartum, providing key insights into microbiome dynamics over the course of a pregnancy. Integrating clinical markers, such as the CL, with microbiome data, enhances the accuracy of PTB predictions. This study has been the first to longitudinally analyze and compare the VMB between term and PTB groups, specifically among pregnant Korean women. This provides valuable insights into microbiome dynamics and potential early indicators for PTB. However, limitations include the study’s focus on a Korean population, limiting generalizability, an inability to establish causality, and the potential exclusion of additional key microbiome markers for PTB.

## 5. Conclusions

This study provides significant insights into the longitudinal changes in the VMB between TB and PTB cases among pregnant Korean women. The findings indicate that the TB group was characterized by a stable, *Lactobacillus*-dominant VMB, particularly enriched with *L. crispatus*, supporting a protective environment. Conversely, the PTB group was associated with increased microbial diversity, marked by elevated levels of potentially pathogenic bacteria such as *P. salivae*, suggesting a correlation between these microbial shifts and the PTB risk. Additionally, CL shortening was correlated with an altered VMB in PTB cases. These findings highlight the potential for integrating VMB profiling and CL measurements as non-invasive, early indicators for a PTB risk, enabling timely interventions to improve pregnancy outcomes. Future research with larger cohorts and diverse populations is essential to validate these findings and enhance the predictive models for PTB. Incorporating VMB and clinical data could facilitate the development of precise, targeted strategies for managing the PTB risk effectively. These findings highlight the exploratory potential of integrating VMB profiling and CL measurements for the early, non-invasive identification of a PTB risk. While our results are promising, they should be interpreted with caution due to the study’s limited sample size and the absence of external validation. Rather than serving as a predictive model per se, this study provides preliminary insights and foundational data that may inform the development of future machine learning-based prediction models for PTB. Furthermore, incorporating VMB and clinical data holds promise for facilitating the development of more precise and targeted strategies to effectively manage the PTB risk.

## Figures and Tables

**Figure 1 medicina-61-00752-f001:**
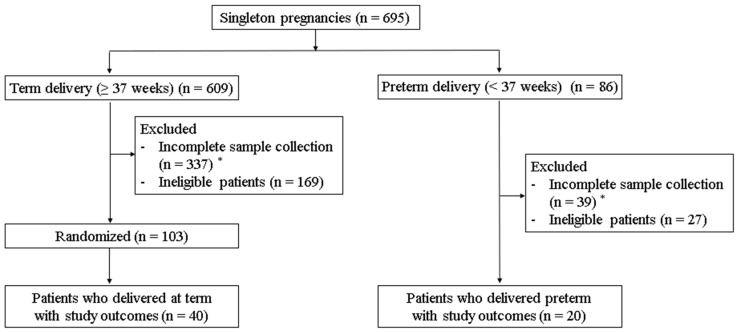
Flow chart of the patient selection. * Only participants who completed sample collection at five time points (term birth group) or three time points (preterm birth group) were included.

**Figure 2 medicina-61-00752-f002:**
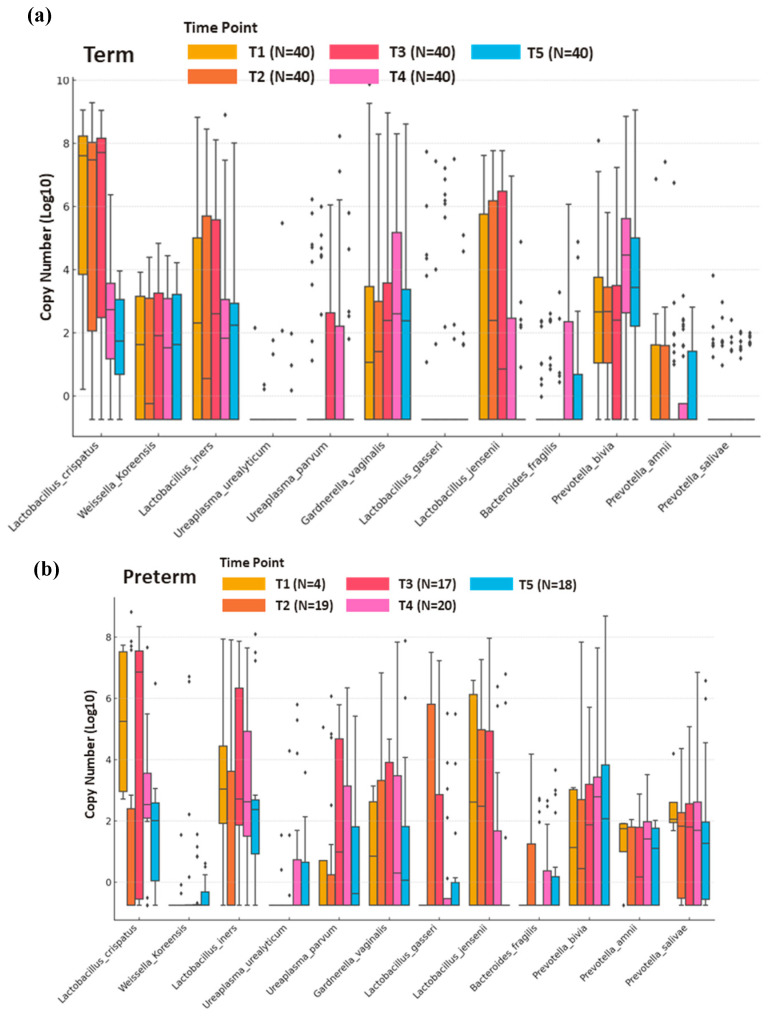
Longitudinal abundance of species comprising the vaginal microbiota in term birth (**a**) and preterm birth groups (**b**) (x-axis represents 12 microbial species, and y-axis indicates microbial copy numbers transformed into Log10 values). T1: first trimester; T2: second trimester; T3: third trimester; P1: 1–2 weeks after delivery; P2: 1–2 months after delivery.

**Figure 3 medicina-61-00752-f003:**
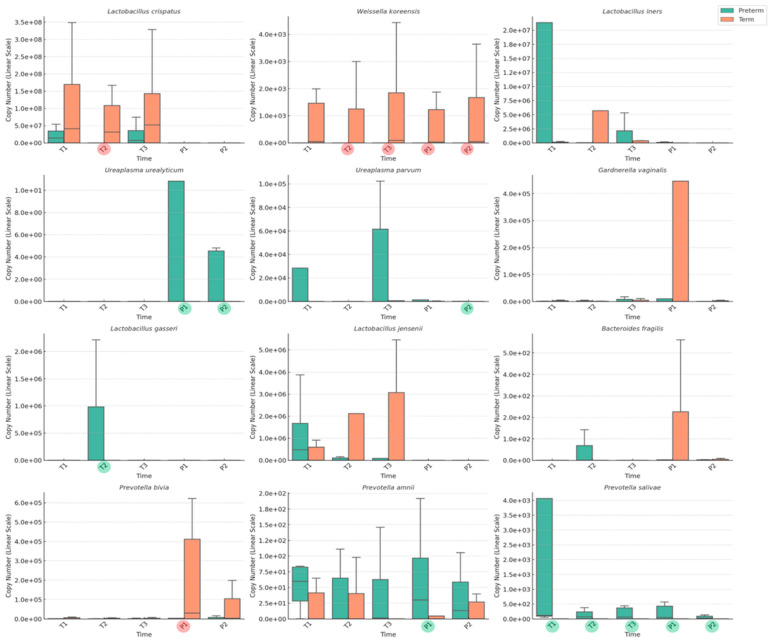
Comparison of vaginal microbiome composition between term and preterm birth groups at five time points (red or green circles above x-axis labels (T1: first trimester; T2: second trimester; T3: third trimester; P1: 1–2 weeks after delivery; P2: 1–2 months after delivery) show statistically significant differences (*p* < 0.05)).

**Table 1 medicina-61-00752-t001:** Sociodemographic data and clinical history.

Characteristics	Term Birth (*n* = 40)	Preterm Birth (*n* = 20)	*p*-Value
Age (years, mean ± SD)	33.0 ± 3.8	34.4 ± 3.8	0.191
Pre-pregnancy BMI (kg/m^2^, mean ± SD)	21.9 ± 3.1	24.8 ± 5.9	0.119
Parity, *n* (%)			
1	20 (50.0)	13 (65.0)	0.296
2	18 (45.0)	5 (25.0)	
3	2 (5.0)	2 (10.0)	
Preterm history, *n* (%)			
Yes	1 (2.5)	1 (5.0)	0.611
No	39 (97.5)	19 (95.0)	
Method of conception, *n* (%)			
Natural pregnancy	32 (80.0)	13 (65.0)	0.206
IVF-ET (ART)	8 (20.0)	7 (35.0)	
Gestational age at sampling (weeks, median (range))			
First trimester	11.6 (11.3–12.5)	12.4 (12.0–12.6)	0.362
Second trimester	24.3 (23.0–25.0)	23.7 (20.6–24.9)	0.994
Third trimester	36.3 (35.4–37.0)	35.1 (34.5–35.3)	0.000 *
Postpartum timing at sampling (days, median (range))			
First sampling	9.0 (8.0–10.8)	9.0 (8.0–11.3)	0.189
Second sampling	43.0 (40.3–45.8)	40.5 (36.3–48.5)	0.294
Use of Lactobacillus supplements, *n* (%)	21 (52.5)	6 (30.0)	0.220
Use of antibiotics or antifungals, *n* (%) ^a^	12 (30.0)	6 (30.0)	1.000
White blood cell count (10^3^/µL, mean ± SD) ^b^	8.8 ± 2.0	10.0 ± 2.6	0.083
C-reactive protein (mg/dL, median (range)) ^b^	0.12 (0.08–0.25)	0.21 (0.06–0.43)	0.393
CL in the second trimester (mm, mean ± SD)	43.0 ± 6.0	38.3 ± 8.1	0.005 *
CL in the third trimester (mm, mean ± SD)	30.9 ± 9.1	19.7 ± 11.2	0.009 *
Mode of delivery			
Vaginal delivery, *n* (%)	14 (35.0)	5 (25.0)	0.624
Cesarean delivery, *n* (%)	26 (65.0)	15 (75.0)	
Gestation age at delivery (weeks, median (range))	38.5 (38.0–39.4)	35.8 (34.2–36.4)	0.000 *
Birth weight (g, mean ± SD)	3232.5 ± 306.8	2421.5 ± 792.2	0.000 *
Sex			
Female, *n* (%)	20 (50.0)	10 (50.0)	1.000
Male, *n* (%)	20 (50.0)	10 (50.0)	
Apgar score at 1 min (median (range))	9.0 (8.0–9.0)	9.0 (7.0–9.0)	0.227
Apgar score at 5 min (median (range))	10.0 (10.0–10.0)	10.0 (8.3–10.0)	0.047 *

SD, standard deviation; BMI, body mass index; ART, assisted reproductive technique; IVF-ET, in vitro fertilization–embryo transfer; CL, cervical length. * *p*-value < 0.05 was considered statistically significant. Continuous variables were expressed as mean ± SD or median (interquartile range) and analyzed using Mann–Whitney U test. Categorical variables were reported as frequencies (percentages) and assessed using Chi-square test or Fisher’s exact test. ^a^ Antibiotics or antifungals were used one week after vaginal sample collection. ^b^ Laboratory exam was performed in second trimester.

**Table 2 medicina-61-00752-t002:** Correlations between species in vaginal microbiome and cervical length of preterm birth group in second and third trimester.

Trimester	Bacteria	Correlation Coefficient ^†^	*p*-Value	Confidence Interval (95%)
T2	*Lactobacillus cripatus*	0.006	0.98	-
T2	*Weissella koreensis*	0.336	0.16	-
T2	*Lactobacillus iners*	0.009	0.97	−0.40~0.42
T2	*Ureaplasma urealyticum*	0.260	0.28	-
T2	*Ureaplasma parvum*	0.187	0.44	-
T2	*Bacteriodes fragilis*	−0.542	0.02 *	−0.79~−0.07
T3	*Lactobacillus crispatus*	0.117	0.75	−0.56~−0.89
T3	*Weissella koreensis*	0.389	0.27	-
T3	*Lactobacillus iners*	0.054	0.88	−0.68~0.86
T3	*Ureaplasma urealyticum*	0.406	0.24	-
T3	*Ureaplasma parvum*	0.109	0.78	−0.62~0.85
T3	*Bacteroides fragilis*	−0.305	0.32	-
T3	*Prevotella bivia*	0.750	0.01 *	0.22~0.95
T3	*Prevotella salivae*	−0.693	0.03 *	−0.94~−0.06

T2, second trimester; T3, third trimester. ^†^ Pearson’s correlation coefficient (rho). * Statistical significance was defined as *p* < 0.05.

## Data Availability

The original contributions presented in this study are included in the article/Appendix A. Further inquiries can be directed to the corresponding author(s).

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
