# Peer review of "A Comparative Longitudinal Study Analyzing Vaginal Microbiota Differences Between Term and Preterm Pregnancies in Korean Women"

_medicina, 2025, doi:10.3390/medicina61040752_

Round 1
Reviewer 1 Report
Comments and Suggestions for Authors
Dear Authors,
This study provides a valuable longitudinal analysis of vaginal microbiota (VMB) differences between term and preterm pregnancies in Korean women. The research is relevant, well-structured, and methodologically sound. The findings contribute to our understanding of the association between microbiota composition, cervical length (CL), and preterm birth (PTB) risk. However, several areas require substantial revision, including methodological clarifications, statistical rigor, and the interpretation of results.
Methodological Clarity
- The selection criteria for study participants require more detail. The manuscript states that 695 women were initially recruited but only 60 were included. The exclusion criteria should be clearly outlined.
- The control of potential confounders (e.g., antibiotic use, probiotics, dietary differences) is insufficiently described. Were participants stratified based on these variables?
- The specific protocol for qPCR needs clarification. While primers and probes are mentioned, how were amplification efficiencies calculated, and was standardization performed across batches?
- The method of microbial DNA extraction is adequately described, but the risk of contamination or degradation during storage and processing should be addressed.
2. Statistical Rigor
- The sample size (20 PTB vs. 40 TB) is relatively small. Was a power analysis performed to justify the sample size?
- Some comparisons lack appropriate statistical adjustments. Were corrections (e.g., Bonferroni or false discovery rate adjustments) made for multiple comparisons in microbiota analyses?
- Correlation coefficients for CL and microbial abundance are presented, but confidence intervals are missing. This should be addressed.
- Some p-values are borderline significant (e.g., p = 0.047 for Apgar score at 5 minutes). Were effect sizes calculated to assess clinical significance?
3. Interpretation and Overgeneralization
- The study suggests that a Lactobacillus-dominant microbiome is protective and that microbial diversity increases PTB risk. However, previous studies show that Lactobacillus iners may be associated with dysbiosis. How do the authors reconcile their findings with existing literature?
- The discussion suggests that W. koreensis might have a protective role, but this claim is speculative. More evidence is needed before drawing such conclusions.
- The study emphasizes the predictive potential of combining microbiota profiling with CL. While promising, this conclusion is premature without external validation or machine learning-based predictive modeling.
4. Figures and Tables
- Figures depicting microbial abundance trends over time need clearer legends and axes labels.
- Tables should include exact p-values rather than thresholds (e.g., p < 0.05).
Reviewer 2 Report
Comments and Suggestions for Authors
VMB is gaining importance and new studies that further the knowledge are most welcome. Despite the limitations this study sheds light on the population specific VMB and shows strong correlation between CL and the presence of Lactobacillus-dominant VMB on one hand annd pathogenic bacteria predominant on the other, and cervical length, a known PTB early marker. It warrants further study for Lactobacillus supplementation, CL and PTB correlation.
Author Response
Comment:
VMB is gaining importance and new studies that further the knowledge are most welcome. Despite the limitations this study sheds light on the population specific VMB and shows strong correlation between CL and the presence of Lactobacillus-dominant VMB on one hand and pathogenic bacteria predominant on the other, and cervical length, a known PTB early marker. It warrants further study for Lactobacillus supplementation, CL and PTB correlation.
Response:
We thank the reviewer for the encouraging and insightful comment. We agree that the relationship between vaginal microbiota and cervical length is of growing interest, especially in the context of predicting and potentially preventing preterm birth. We appreciate the acknowledgment of our findings that highlight a population-specific association between Lactobacillus-dominant or pathogenic microbiota and cervical length. As suggested, we have noted in the discussion the importance of further studies investigating the potential role of Lactobacillus supplementation in modulating cervical length and reducing preterm birth risk.
Reviewer 3 Report
Comments and Suggestions for Authors
This study aims to analyze VMB differences between term and preterm pregnancies in Korean women and their correlation with cervical 22 length (CL) and to see if microbiome influences the premature birth. It compares the microbiome of pregnant Korean women and found a correlation between the microbiome and term and preterm pregnancies.
I think the study is original and relevant to the field. If further research studies will confirm these findings than the obstetricians will have the possibility to influence the microbiome and to prevent some of the premature births. The recommendation for improving the study's methodology is to introduce the exclusion criteria more accurately.
Overall, the paper is very well written and very interesting. In the exclusion criteria from the study is mentioned medical conditions and I think it will be interesting to introduce a table with these medical conditions.
Author Response
Comment:
This study aims to analyze VMB differences between term and preterm pregnancies in Korean women and their correlation with cervical length (CL) and to see if microbiome influences the premature birth. It compares the microbiome of pregnant Korean women and found a correlation between the microbiome and term and preterm pregnancies. I think the study is original and relevant to the field. If further research studies will confirm these findings than the obstetricians will have the possibility to influence the microbiome and to prevent some of the premature births. The recommendation for improving the study's methodology is to introduce the exclusion criteria more accurately. Overall, the paper is very well written and very interesting. In the exclusion criteria from the study is mentioned medical conditions and I think it will be interesting to introduce a table with these medical conditions.
Response:
We sincerely appreciate the reviewer’s positive evaluation of the originality and clinical relevance of our study. We agree that confirmation of our findings in larger-scale studies could provide obstetricians with new avenues for intervention in preventing preterm births. Regarding the reviewer’s helpful suggestion to clarify the exclusion criteria, we have revised the methods section (lines 60-71) to describe these criteria in more detail. We believe this addition improves the transparency and reproducibility of our methodology. Thank you again for your valuable comments.
Reviewer 4 Report
Comments and Suggestions for Authors
This is an interesting study. However, it should be pointed that it is not known if interventions to favorably alter the vaginal microbiota will finally lower the risk of PTB.
Author Response
Comment:
This is an interesting study. However, it should be pointed that it is not known if interventions to favorably alter the vaginal microbiota will finally lower the risk of PTB.
Response:
We appreciate the reviewer’s thoughtful comment. We acknowledge that while our study suggests an association between vaginal microbiota composition and cervical length in term and preterm pregnancies, it does not establish a causal link between microbiota modification and reduced PTB risk. Further interventional studies are necessary to determine whether targeted microbiota modification strategies, such as Lactobacillus supplementation, can effectively lower the incidence of PTB. We have noted this limitation in the discussion section to clarify the need for future research in this area.
Round 2
Reviewer 1 Report
Comments and Suggestions for Authors
Dear Authors
thanks for this revised version of the manuscript.
You addressed all the reviewer's concerns.